# Tree-based Ensemble Learning for Out-of-distribution Detection

## Abstract

Being able to successfully determine whether the testing samples has similar distribution as the training samples is a fundamental question to address before we can safely deploy most of the machine learning models into practice. In this paper, we propose TOOD detection, a simple yet effective tree-based out-of-distribution (TOOD) detection mechanism to determine if a set of unseen samples will have similar distribution as of the training samples. The TOOD detection mechanism is based on computing pairwise hamming distance of testing samples' tree embeddings, which are obtained by fitting a tree-based ensemble model through in-distribution training samples. Our approach is interpretable and robust for its tree-based nature. Furthermore, our approach is efficient, flexible to various machine learning tasks, and can be easily generalized to unsupervised setting. Extensive experiments are conducted to show the proposed method outperforms other state-of-the-art out-of-distribution detection methods in distinguishing the in-distribution from out-of-distribution on various tabular, image, and text data.

## 1 Introduction

A fundamental assumption which assures any machine learning model associated with a training and testing phase to succeed is that the training and testing data should follow a similar distribution. However, this assumption may not be valid in practice, in which case, it is called out-of-distribution (OOD). Out-of-distribution detection is a fundamental and crucial task in many disciplines such as health science, engineering, geophysics, etc,. For example, if we need to make drug recommendation to new patients, it is critical to make sure the features of new patients are similar to the patients whose consequences of taking the drug are known.

There has been a plethora of research so far which address on OOD detection (Hendrycks & Gimpel, 2017; Hendrycks et al., 2019; Hsu et al., 2020; Lee et al., 2018; Liang et al., 2018; Lakshminarayanan et al., 2017; Mohseni et al., 2020). Starting from (Hendrycks & Gimpel, 2017), which introduces a common baseline for OOD detection based on softmax probability, followed by more recent works such as energy based approach (Liu et al., 2020; Grathwohl et al., 2020), likelihood ratio based approach (Ren et al., 2019; Serrà et al., 2020), and generative model based approach (Choi & Jang, 2018), just to name a few. More recently, a comprehensive survey about OOD detection is given in (Yang & Liu, 2021).

As artificial neural network has so far been the most popular model for various types of tasks, almost all of the aforementioned OOD detection methods are neural network based, which generally means that a set of neural network parameters are learned in the training phase by using the training samples, and then a score or metric will be imposed on the unseen testing samples to determine whether the testing samples are similar or not to the distribution of training samples. Despite its partial success in achieving such a goal, due to the neural network's black-box feature, the rationale behind such approaches are not very intuitive.

In addition, the neural network based models are often vulnerable to adversarial attack (Carlini & Wagner, 2017; Goodfellow et al., 2015; Kurakin et al., 2018; Moosavi-Dezfooli et al., 2016), which makes such model less reliable in practice. Researchers have also been studying on using different metrics such as ODIN score (Liang et al., 2018), generalized ODIN score (Hsu et al., 2020), Mahalanobis distance (Lee et al., 2018) to improve the OOD detection performance. However, these approaches are often computationally expensive therefore can not be easily applied in practice.

Furthermore, neural network based models are usually restricted to the supervised setting and often require fine-tuning in order to make it works well for specific tasks and datasets. For example, previous methods have used synthetic data or unlabeled data (Hendrycks et al., 2019; Lee et al., 2017) as auxiliary OOD training data, which enables explicitly regularization of the model through fine-tuning and leads to low confidence scores on anomalous examples. The work (Mohseni et al., 2020) investigated training methods involving the inclusion of extra background classes to improve OOD scoring. The work (Chen et al., 2021) proposed informative outlier mining by selectively training on auxiliary OOD data that induces uncertain OOD scores, which improves the OOD detection performance on both clean and perturbed adversarial OOD inputs.

In order to deal with these disadvantages, we propose TOOD detection, a simple yet effective tree-based OOD detection method. Inherited from the characteristics of tree-based machine learning models, the main advantages of TOOD detection are the following four aspects: *Interpretability*, *Robustness*, *Flexibility*, and *Efficiency*. It is interpretable in that there is no black-box feature involved in the proposed method and it can be easily interpreted just like a decision tree or random forest. It is robust in that the method will give stable outputs for perturbed inputs, e.g., an adversarial attack for images. It is efficient in that the method is easy to train and is often faster than neural network based methods. It is flexible in that the model requires little or no fine-tuning of the model's parameters, can handle various machine learning tasks with different types of input data, and can be easily generalized in an unsupervised way.

The rest of paper is structured as follows. Section 2 gives a general overview of the tree-based ensemble learning methods. Section 3 proposes a specific tree-based learning procedure for out-of-distribution detection and provides its intuition and rationale. We give a detailed analysis of the correctness for the proposed method in Section 4 and present extensive experimental results in Section 5. In Section 6, we discuss several factors which may have impact on the results of our method and how to generalize our method to the unsupervised setting. Finally, we conclude the paper and point out some potential future research directions in Section 7.

## 2 BACKGROUND

Tree-based ensemble learning is a traditional but popular machine learning model for classification and regression tasks. It has merits in handling different types of input features, requiring little or no data preprocessing, and being easy to train and interpret. Decision tree (Quinlan, 1979; Breiman et al., 1984) and random forest (Ho, 1995; 1998; Breiman, 2001) are two classic examples of tree-based learning models. There are also other types of tree-based learning models such as extremely randomized tree (Geurts et al., 2006) and isolation forest (Liu et al., 2008).

### 2.1 DECISION TREE AND RANDOM FOREST

A decision tree is a graphical representation of a decision-making process. It is one of the most popular supervised learning models in machine learning. It is a tree-like structure consisting of nodes and branches which represent decisions and their possible consequences. In a decision tree, each internal node represents a test on an attribute, each branch represents the outcome of the test, and each leaf node represents a class label or a decision. It starts with a root node that represents the entire dataset and recursively partitions the dataset into smaller subsets based on the values of the input attributes, until a stopping criterion is met or a decision is made.

Random forest is an ensemble of multiple decision trees. It builds many decision trees and combines the output of all the trees as weaker learners to form a strong learner. In a random forest, each decision tree is usually fitted on a boostrapped subsampled data and subsampled input features. The way to find the best split point among these features is based on some criterion such as Gini impurity or information gain. Once the model is fitted, the outputs of the individual trees are combined through a voting system to make a final prediction.

### 2.2 EXTREMELY RANDOMIZED TREE

Extremely randomized tree has a similar theme as random forest. However, instead of finding the best split point of each feature such as in random forest, it picks a random threshold to make each

split. The extra randomness from the random thresholds can lead to trees with higher bias but usually with reduced variance when compared to random forest. Given the randomness in picking thresholds, the training process can often be faster than random forest, as it avoids the exhaustive search for the best split point. We refer to (Geurts et al., 2006) for a more detailed discussion of extremely randomized tree.

## 3 TREE-BASED OUT-OF-DISTRIBUTION DETECTION

Despite the success of tree-based ensemble models in tackling tasks such as classification and regression, their power for other tasks are still under-explored. Recall that OOD detection is the process of identifying data samples that belong to a different distribution than the one used to train a machine learning model. Let us explain the intuition of our method for OOD detection in the following subsections.

### 3.1 TREE EMBEDDING

Our idea is based on the following observation. On the left side of Figure 1, there are four training samples and they are separated by a horizontal line $y = y_1$ and a vertical line $x = x_1$. In the training phase, a tree-based ensemble learning model, e.g., random forest, is learned. By choosing appropriate hyperparameters, the model will build some classification trees. For convenience, let us assume there are two trees being built during training, as shown on the right side of Figure 1.

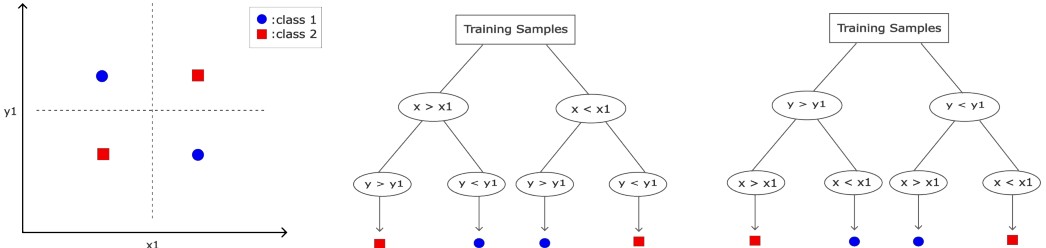

Figure 1: Training samples for tree-based ensemble learning

Using these two trees as references, the lower left sample corresponds to the *fourth* leaf node in the first tree and *fourth* leaf node in the second tree, the upper left sample corresponds to the *third* leaf in the first tree and *second* leaf node in the second tree, the upper right sample corresponds to the *first* leaf node in the first tree and *first* leaf node in the second tree, the lower right sample corresponds to the *second* leaf node in the first tree and *third* leaf node in the second tree. In other words, for each sample, we use a vector to indicate which leaf node the sample will reach for all trees. This gives us a tree embedding in the feature space.

So far, we have obtained row vectors $[4, 4]$, $[3, 2]$, $[1, 1]$, $[2, 3]$ as the tree embedding for the lower left sample, upper left sample, upper right sample, lower right sample, respectively. Let us call them the first, second, third, fourth samples. More conveniently, we can put these vectors into a tree embedding matrix as

$$\begin{bmatrix} 4 & 4 \\ 3 & 2 \\ 1 & 1 \\ 2 & 3 \end{bmatrix}. \tag{1}$$

Recall that the hamming distance $d$ between two vectors $\boldsymbol{x}_1, \boldsymbol{x}_2 \in \mathbb{R}^n$ is defined as the normalized number of components where these two vectors differ, i.e., $d(\boldsymbol{x}_1, \boldsymbol{x}_2) := \frac{1}{n}\|\boldsymbol{x}_1 - \boldsymbol{x}_2\|_{\ell_0}$. We calculate the average pairwise hamming distance (APHD) for each fixed training sample (each row of the embedding matrix) against other training samples. For example, for the first sample in our case, we have $d([4, 4], [3, 2]) = 1$, $d([4, 4], [1, 1]) = 1$, $d([4, 4], [2, 3]) = 1$. The APHD for the first sample is defined as the average of these three values, which equals to $1$. Similarly, the APHD of the second, third, and fourth samples are all equal to $1$.

## 3.2 OUT-OF-DISTRIBUTION DETECTION

Once we have the fitted tree-based ensemble model, the unlabeled testing samples will be feed into the model and generate a tree embedding for each individual testing sample.

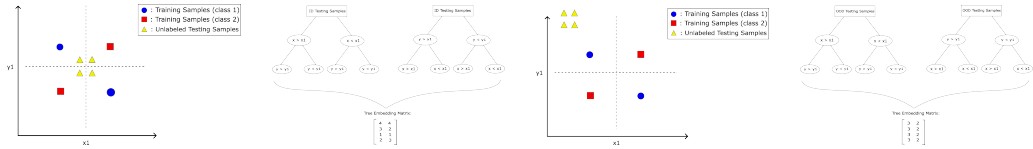

Figure 2: Tree-based ensemble learning for in-distribution (left panel) and out-of-distribution (right panel)

We can calculate the APHD for each testing sample against other testing samples the same way as we did for the training samples. These AHPD values can be used as an indicator to determine whether the whole testing data is more likely from an in-distribution or out-of-distribution data. For example, in the left panel of Figure 2, the APHD values for the four testing samples equal to $1, 1, 1, 1$, in the right panel of Figure 2, the APHD values for the four testing samples equal to $0, 0, 0, 0$. Therefore, we can choose some threshold value, say $0.5$, to separate the in-distribution from out-of-distribution data. Let us summarize the idea into Algorithm 1.

---

**Algorithm 1** Tree-based Out-of-distribution Detection (TOOD Detection)

---

1: **Input:** Training samples $\{\boldsymbol{x}_i^{train}\}_{i=1}^N \subset \mathcal{D}_{train}$, testing samples $\{\boldsymbol{x}_i^{test}\}_{i=1}^M \subset \mathcal{D}_{test}$, class labels $y_i^{train} \in \{1, 2, \cdots, K\}$ for $i = 1, \cdots, N$.
2: Fit a random forest or extremely randomized tree model based on $\{\boldsymbol{x}_i^{train}\}_{i=1}^N$ and $\{y_i^{train}\}_{i=1}^N$ to obtain the tree embedding feature map.
3: Feed testing samples $\{\boldsymbol{x}_i^{test}\}_{i=1}^M \subset \mathcal{D}_{test}$ into the model to obtain their tree embedding vectors and form the tree embedding matrix.
4: For each testing sample (a row in the embedding matrix), calculate its APHD against all the other testing samples.
5: **Output:** The APHD values for all testing samples.

---

Intuitively, in-distribution samples will have larger APHD values since samples from in-distribution are more likely to be separated by the decision boundaries obtained from training if the samples on the opposite sides of each decision boundary have different labels, which results in a larger pairwise hamming distance. On the other hand, out-of-distribution samples will have smaller APHD values since they are more likely to appear only on one side of each decision boundary and hence are less likely to be separated by the those decision boundaries. Therefore, we can use APHD values to distinguish in-distribution data from out-of-distribution data. Let us give a more detailed analysis in the next section.

## 4 ANALYSIS

For convenience, let us assume all the decision trees considered from now on are binary, the number of samples for each leaf node equals to $1$, and all the trees are pruned to be minimal. Let us also assume all the constraints in the decision nodes are hyperplanes which are orthogonal to some axes (e.g., $x > x_1$ or $y > y_1$, but not $x + y > x_1 + y_1$). Suppose our data of interest lies in some underlying manifold $\mathcal{M}$ of certain dimension. From the tree-based ensemble learning model which is obtained through training samples, we can define the tree embedding as $T = (T_1, \cdots, T_L) : \mathcal{M} \to \mathbb{R}^L$, where $T_\ell, \ell = 1, \cdots, L$, is the tree embedding for the $\ell$-th tree, $L$ is the total number of trees.

Let $\mathcal{H}_\ell := \{H_1, H_2, \cdots, H_m\}$ be the set of all decision regions when growing the $\ell$-th tree. For example in Figure 1, we have $\mathcal{H}_1 = \mathcal{H}_2 = \{H_1, H_2, H_3, H_4\}$ where $H_1 = \{(x, y) : x > x_1, y > y_1\}$, $H_2 = \{(x, y) : x > x_1, y < y_1\}$, $H_3 = \{(x, y) : x < x_1, y > y_1\}$, $H_4 = \{(x, y) : x < x_1, y < y_1\}$. It is worthwhile to note the decision regions from a same tree are mutually exclusive. The following results can be established and we defer all the proofs to the appendix.

**Lemma 1** *Let $\mathcal{H}_\ell$ be the set of all decision regions obtained from the training samples when growing the $\ell$-th tree. For any $\boldsymbol{x}_1, \boldsymbol{x}_2 \in \mathcal{M}$, we have*

$$d(T_\ell(\boldsymbol{x}_1), T_\ell(\boldsymbol{x}_2)) = \begin{cases} 0 & if \quad \boldsymbol{x}_1, \boldsymbol{x}_2 \in H \quad for\ some \quad H \in \mathcal{H}_\ell, \\ 1 & otherwise. \end{cases} \tag{2}$$

Now we are ready to show the correctness of our approach. In other words, we want to show the APHD values will be significant different for data coming from in-distribution versus data coming from out-of-distribution. Suppose all the training and testing samples are of dimension $n$, let us first show a simple case when $\mathcal{D}_{train}$ and $\mathcal{D}_{test}$ are easily separated. Here we use the notation $Conv(supp(A))$ to indicate the convex hull of the support of set $A$.

**Theorem 1** *Suppose $Conv(supp(\mathcal{D}_{test})) \cap Conv(supp(\mathcal{D}_{train})) = \emptyset$. Then for any pair of samples $\boldsymbol{x}_i, \boldsymbol{x}_j \in \mathcal{D}_{test}$, we have*

$$d(T_\ell(\boldsymbol{x}_i), T_\ell(\boldsymbol{x}_j)) = 0. \tag{3}$$

Next, let us see the expected hamming distance for any pair of samples from a distribution that is same as $\mathcal{D}_{train}$ is away from 0, and hence the expected AHPD values are away from 0. Therefore, it is reasonable to choose some threshold in order to distinguish in-distribution data from out-of-distribution data.

**Theorem 2** *Suppose $\mathcal{H}_\ell$ divides the data manifold in $\mathbb{R}^n$ into $K = O(k^n)$ different decision regions (where $k$ is roughly the number divided pieces for each dimension) based on training samples. If $\mathcal{D}_{test}$ follows the same distribution as $\mathcal{D}_{train}$ and suppose all testing samples have equal probability to occur across all $K$ decision regions. Then for any $\boldsymbol{x}_i, \boldsymbol{x}_j \in \mathcal{D}_{test}$, their expected tree embedded hamming distance is*

$$\mathbb{E}[d(T_\ell(\boldsymbol{x}_i), T_\ell(\boldsymbol{x}_j))] = \frac{K - 1}{K}. \tag{4}$$

*In particular, we have $\mathbb{E}[d(T_\ell(\boldsymbol{x}_i), T_\ell(\boldsymbol{x}_j))] \to 1$ as $K \to \infty$.*

**Remark 1** *As we will see in Figure 3 from the experiments, the AHPD values for real image datasets are usually close to 1, that is because the distribution of images are so complicated that the tree-based model divides the entire spaces into enormous decision regions. It is also intuitive that $K$ will grow as the number of training samples increases, this agrees with our observation in the top row of Figure 4.*

However, data in reality can be complicated and it is often not the case that $Conv(supp(\mathcal{D}_{test})) \cap Conv(supp(\mathcal{D}_{train})) = \emptyset$. In general, it is not easy to estimate the expected tree embedded hamming distance unless the data has certain nice distribution. For data with uniform distribution, we can establish the following result.

**Theorem 3** *Suppose $\mathcal{D}_{train}$ follows uniform distribution on $[a_1, b_1]^n$ and $\mathcal{D}_{test}$ follows uniform distribution on $[a_2, b_1 + a_2 - a_1]^n$ where $a_1 \leq a_2 \leq b_1$. If for all $\ell = 1, \cdots, L$, $\mathcal{H}_\ell$ divides the data manifold in $\mathbb{R}^n$ into exponentially many ($k \gg 1$ for each dimension) decision regions based on training samples from $\mathcal{D}_{train}$. Then for any pair of testing samples $\boldsymbol{x}_i, \boldsymbol{x}_j \in \mathcal{D}_{test}$, we have*

$$\mathbb{E}[d(T_\ell(\boldsymbol{x}_i), T_\ell(\boldsymbol{x}_j))] = 1 - \left(\frac{a_2 - a_1}{b_1 - a_1}\right)^{2n}. \tag{5}$$

*In particular, if $a_2 = b_1$, then $\mathbb{E}[d(T_\ell(\boldsymbol{x}_i), T_\ell(\boldsymbol{x}_j))] = 0$. If $a_2 = a_1$, then $\mathbb{E}[d(T_\ell(\boldsymbol{x}_i), T_\ell(\boldsymbol{x}_j))] = 1$. These agree with the results in Theorem 1 and 2.*

**Remark 2** *We can see that as data dimension increases, the expected pairwise hamming distance also increases. This agrees with our observation in Figure 7 that the distribution becomes more indistinguishable as the dimension increases. This phenomenon also reflects the issue of curse of dimensionality. However, in practice, we can reduce the data dimension by extracting the latent features via autoencoders, as demonstrated in our experiments for image data.*

Now we can establish the consistency result of pairwise hamming distance for ensemble of multiple trees by using Hoeffding's inequality.

**Theorem 4** *Suppose the total number of trees being built in the ensemble model equals to L. Under the same assumptions as Theorem 3, the tree embedded pairwise hamming distance for any pair of testing samples $\boldsymbol{x}_i, \boldsymbol{x}_j \in \mathcal{D}_{test}$ satisfies*

$$\mathbb{E}[d(T(\boldsymbol{x}_i), T(\boldsymbol{x}_j))] = 1 - \left(\frac{a_2 - a_1}{b_1 - a_1}\right)^{2n}, \tag{6}$$

*and*

$$P\Big(\Big|d(T(\boldsymbol{x}_i), T(\boldsymbol{x}_j)) - \mathbb{E}[d(T(\boldsymbol{x}_i), T(\boldsymbol{x}_j))]\Big| \geq t\Big) \leq 2\exp(-2t^2 L). \tag{7}$$

**Remark 3** *The significance of Theorem 4 is that it is an instance-based result, which is more desirable than sample-based result, and can be applied to a more general scenarios in practice. In the case of there are only very few testing samples, it is still applicable as long as there are enough training samples.*

## 5 EXPERIMENTS

In this section, we demonstrate the effectiveness of TOOD detection method by evaluating it on several benchmark datasets and comparing it with other state-of-the-art OOD detection methods. We choose to only show the sample-based OOD detection results for making the comparison easier with other methods. To show that our approach is broadly applicable to various machine learning tasks, we perform experiments on tabular data analysis, computer vision, and natural language processing tasks. We present the mean values for all the experiments. The full results in the *mean $\pm$ std* form are provided in the appendix. For all the means and standard deviations presented in this section and in appendix, values are percentages, and are rounded so that $99.95\%$ rounds to $100\%$ and $0.049\%$ rounds to $0\%$.

For simulated and tabular data, we directly use the original training samples as input for the tree-based model. For image or text data, we impose an autoencoder or word embedding to extract latent features and use latent features as input for our tree-based model. More details of the datasets usage, experimental procedures, hyperparameters, and evaluation are provided in the appendix. For reproducibility purpose, we make our code available in the supplementary material.

### 5.1 PRELIMINARY RESULTS

Table 1: AUROC values of TOOD detection results. Rows are in-distribution datasets, columns are out-of-distribution datasets.

|              | BankMarketing | Diabetes130US | Electricity | Gaussian | Uniform |
|--------------|---------------|---------------|-------------|----------|---------|
| BankMarketing | -            | 100           | 100         | 100      | 99.9    |
| Diabetes130US | 100          | -             | 100         | 100      | 100     |
| Electricity   | 99.9         | 100           | -           | 99.9     | 97.4    |

Table 2: AUROC values of TOOD detection results. Rows are in-distribution datasets, columns are out-of-distribution datasets.

|              | IMDB | AGNEWS | Amazon | YahooAnswers | Yelp |
|--------------|------|--------|--------|--------------|------|
| IMDB         | -    | 100    | 96.2   | 99.4         | 99.8 |
| AGNEWS       | 100  | -      | 99.7   | 98.8         | 100  |
| Amazon       | 98.3 | 99.5   | -      | 97.4         | 98.6 |
| YahooAnswers | 99.6 | 98.9   | 93.7   | -            | 98.0 |
| Yelp         | 99.8 | 100    | 98.6   | 99.5         | -    |

For computer vision tasks, we can see in Table 3 that TOOD detection performs very well on MNIST-like datasets. It is worth noting that QMNIST is also a hand-written digits dataset and

is very similar to MNIST. This can be seen through its AUROC, AUPR, FPR95 value that QMNIST is indistinguishable from MNIST but can easily be distinguished from FashionMNIST, while other MNIST-like data are both distinguishable from MNIST and FashionMNIST. This shows that TOOD detection is indeed able to tell whether there are intrinsic differences between images.

Table 3: TOOD detection results for MNIST and FashionMNIST datasets. Expanded results are provided in the appendix.

| $\mathcal{D}_{in}$ | $\mathcal{D}_{out}$ | AUROC ↑ | AUPR ↑ | FPR95 ↓ |
|---|---|---|---|---|
| **MNIST / FashionMNIST** | MedMNIST | 99.9 / 99.9 | 99.9 / 99.8 | 0.23 / 0.37 |
| | KMNIST | 95.8 / 99.2 | 96.7 / 99.1 | 20.2 / 4.20 |
| | QMNIST | 48.7 / 100 | 49.6 / 100 | 96.4 / 0 |

## 5.2 COMPARISON WITH STATE-OF-THE-ARTS

We also summarize the results of TOOD detection and several other state-the-of-art OOD detection for computer vision and natural language tasks in Table 4, 5, and 6. Our method is shown to have comparable or favorable performance to the state-of-the-art results. Especially for the 20Newsgroups dataset, it outperforms other methods by a quite large margin.

To gain some further insights, we perform an extra experiment on STL-10 (a dataset somewhat similar to CIFAR-10) as the testing data while using CIFAR-10 as the in-distribution training data. The AUROC, AUPR, FPR95 for this experiment equals to 83.2%, 78.6%, 46.2%, respectively. This again shows that our method is indeed able to tell whether there are intrinsic differences between images. The expanded experimental results on each individual OOD dataset are provided in the appendix.

Table 4: Comparison with the basline MSP (Hendrycks & Gimpel, 2017). The in-distribution dataset is MNIST.

| $\mathcal{D}_{out}$ | AUROC ↑ | AUPR ↑ |
|---|---|---|
| | MSP / TOOD (ours) | |
| Omniglot | 96 / **100** | 97 / **100** |
| NotMNIST | 87 / **100** | 88 / **100** |
| CIFAR10-bw | 98 / **100** | 98 / **100** |
| Gaussian | 90 / **100** | 90 / **100** |
| Uniform | 99 / **100** | 99 / **100** |

Table 5: Comparison with other methods whose results are based on WideResNet. The results are averaged over six OOD datasets: SVHN, Texture, Places365, iSUN, LSUN-Crop, LSUN-Resize.

| $\mathcal{D}_{in}$ | OOD detection Methods | AUROC ↑ | AUPR ↑ | PFR95 ↓ |
|---|---|---|---|---|
| **CIFAR-10** | MSP (Hendrycks & Gimpel, 2017) | 90.1 | 97.9 | 51.0 |
| | ODIN (Liang et al., 2018) | 91.1 | 97.6 | 35.7 |
| | Mahalanobis (Lee et al., 2018) | 93.3 | 98.5 | 37.1 |
| | OE (Hendrycks et al., 2019) | 98.3 | 99.6 | 8.53 |
| | Energy score (Liu et al., 2020) | 91.9 | 97.8 | 33.0 |
| | Energy fine tuning (Liu et al., 2020) | 98.9 | **99.8** | **3.32** |
| | TOOD (ours) | **99.1** | 98.8 | 3.93 |
| **CIFAR-100** | MSP (Hendrycks & Gimpel, 2017) | 75.5 | 93.9 | 80.4 |
| | ODIN (Liang et al., 2018) | 77.4 | 94.2 | 74.6 |
| | Mahalanobis (Lee et al., 2018) | 84.1 | 95.9 | 54.0 |
| | OE (Hendrycks et al., 2019) | 85.2 | 96.4 | 58.1 |
| | Energy score (Liu et al., 2020) | 79.6 | 94.9 | 73.5 |
| | Energy fine tuning (Liu et al., 2020) | 88.5 | **97.1** | 47.6 |
| | TOOD (ours) | **94.2** | 94.4 | **33.4** |

Table 6: Comparison with other out-of-distribution detection methods MSP (Hendrycks & Gimpel, 2017), OE (Hendrycks et al., 2019), PnPOOD (Rawat et al., 2021) on 20Newsgroups dataset.

| $\mathcal{D}_{in}$ | $\mathcal{D}_{out}$ | AUROC ↑ | AUPR ↑ | FPR90 ↓ |
|---|---|---|---|---|
| | | MSP / OE / PnPOOD / TOOD (ours) | | |
| Computer | Sports | 62 / 90 / 92 / **99.7** | 23 / 64 / 65 / **98.9** | 72 / 26 / 18 / **0.4** |
| | Politics | 63 / 92 / 93 / **99.2** | 24 / 67 / 68 / **97.2** | 72 / 15 / 11 / **1.1** |
| Sports | Computer | 63 / 82 / 89 / **99.8** | 23 / 35 / 51 / **99.7** | 71 / 32 / 22 / **0.3** |
| | Politics | 61 / 82 / 87 / **98.8** | 21 / 36 / 51 / **97.0** | 76 / 30 / 24 / **2.3** |
| Politics | Computer | 67 / 91 / 92 / **99.8** | 25 / 64 / 60 / **99.7** | 61 / 24 / 20 / **0.6** |
| | Sports | 67 / 85 / 88 / **99.6** | 25 / 53 / 56 / **99.5** | 63 / 42 / 34 / **0.9** |

## 6 DISCUSSIONS

Let us have a more detailed discussion on how the data size and dimension will affect the APHD values of the in-distribution and out-of-distribution samples. We will also see that our method is robust to adversarial attack such as FGSM (Goodfellow et al., 2015) and can be generalized to the unsupervised setting.

**Effect of data size on APHD values and TOOD detection results**  We have shown in Theorem 2 that the APHD values for in-distribution dataset increases as the number of decision regions increases. This can also be seen through Figure 3 that the decision boundaries of an in-distribution data such as CIFAR-10 are complicated enough so that its APHD values are close to 1. Meanwhile, as the number of decision regions increase, the APHD values for out-of-distribution data will also increase. For fixed dataset, one way to increase the APHD values is to increase the number of training samples, hence potentially increase the number of decision regions. Top row of Figure 4 shows that the more in-distribution training samples there are, the larger APHD values we get for both in-distribution and out-of-distribution data.

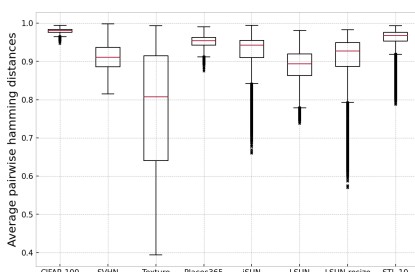

Figure 3: Boxplot of AHPD values when CIFAR-10 is the in-distribution dataset.

**Effect of data dimension on APHD values and TOOD detection results**  To further investigate on how the data dimension will affect the APHD values and TOOD detection results, we design an auxiliary experiment with three simulated datasets which consist of point clouds in $\mathbb{R}^n$ whose two dimensional projection look like three particular geometric shapes: circles, lines, squares. Their two dimensional projections are shown in Figure 6 in the appendix. We use circles as in-distribution data and the other two as out-of-distribution data, the dimensions are increased as $n = 5, 10, 30, 100$. The APHD values for each dimension is shown in Figure 7. We can see that higher dimension complicates the data distribution and makes in-distribution and out-of-distribution data indistinguishable.

**Robustness**  The tree-based ensemble learning method is known for its robustness against noise. We conduct another experiment to validate this point. We apply FGSM attack on several testing image data while using CIFAR-10 and CIFAR-100 as the training (in-distribution) data respectively. Figure 5 shows that the TOOD detection results under FGSM attack do not compromise too much compared with the case without attack.

**Towards unsupervised learning**  One notable feature of the TOOD detection method is that it uses information from training labels to divide the space into separate decision regions. It turns out

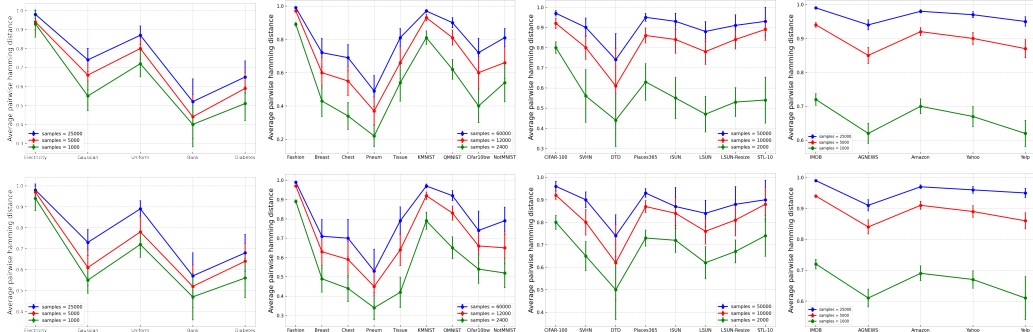

Figure 4: APHD values for different training sample sizes (Top row: with original labels. Bottom row: with randomly shuffled labels). From left to right: Electricity (in-distribution) vs Others (out-of-distribution); FashionMNIST (in-distribution) vs Others (out-of-distribution); CIFAR-100 (in-distribution) vs Others (out-of-distribution); IMDB (in-distribution) vs Others (out-of-distribution)

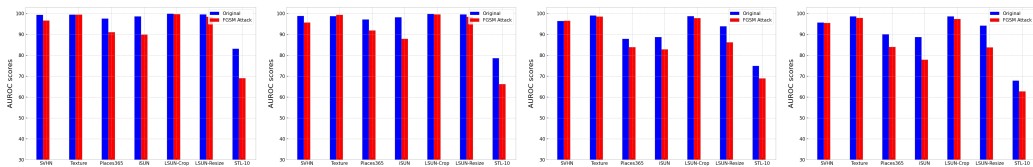

Figure 5: AUROC and AUPR scores of original images and images under FGSM attack for CIFAR-10 (first and second subplots) and CIFAR-100 (third and fourth subplots) as in-distribution data

that the labels themselves do not carry too much information for the purpose of out-of-distrituion detection. In other words, we can randomly shuffle the training labels and then fit a tree-based ensemble model in the same fashion. In this way, the classification results are not as good as original but the TOOD detection results are as good as if without shuffling the labels. This is likely because TOOD detection uses pairwise hamming distance as the criterion to determine the OOD samples. Intuitively speaking, even though the labels are randomly shuffled, as long as each sample itself and its neighbours are of different labels, the tree-based model will still be able to divide the space into different decision regions as if without randomly shuffling the labels, and hence the pairwise hamming distance will not be affected. This can be seen through the bottom row of Figure 4, where each of the subplots looks similar to its top counterpart.

# 7 Conclusions and Future Directions

In this work, we proposed TOOD detection, an interpretable, robust, efficient, and flexible tree-based ensemble learning approach for out-of-distribution detection. The proposed method works well on various datasets and achieves comparable and favorable results than other state-of-the-art out-of-distribution detection methods. TOOD detection is easy to train, requires little or no parameters fine-tuning and is shown to be robust to adversarial attack, which most of the neural network approaches are lack of. One potential future research direction can be developed on investigating how to use feature engineering and dimension reduction techniques to find a better embedding space so that the tree-based method for out-of-distribution detection will be more effective. Another direction is about how to learn the inverse map of the proposed tree embedding. If we were able to learn its inverse map, then we can use the inverse map to easily generate data which is likely to be the data coming from the same distribution as original data. This perspective can be useful in potentially developing a new scheme for generative model. In summary, we hope this work will bring people's interests into this new perspective of out-of-distribution detection mechanism and potentially stimulate more research towards this direction in the future.

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

## A   DATASETS

The experimental datasets are divided into four different categories: simulated data, tabular data, image data, and text data. For all in-distribution datasets, we make use of both the training and testing part of the datasets. For all out-of-distribution (OOD) datasts, we only use the testing part of the datasets. If the size of testing part of an OOD dataset is too small, we will use its training part to substitute the testing part. If a dataset only contains a single part as a whole, we will apply standard training and testing split to seperate it into two parts. A brief summary of the datasets are given below.

**Simulated data**   We perform evaluation on three simulated high dimensional point clouds whose 2D projections look like circles, lines, squares. Their projections are shown in Figure 6.

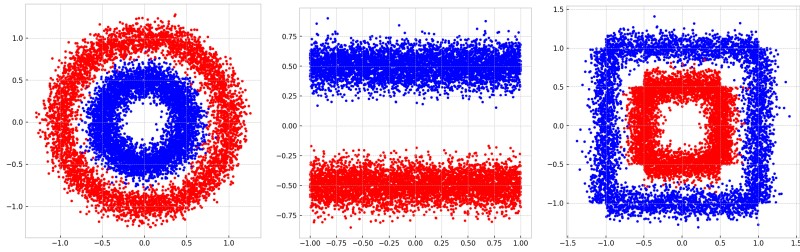

Figure 6: 2D projection of simulated high dimensional point clouds with three particular shapes

**Tabular data**   We perform evaluation on three tabular datasets: BankMarketing, Diabetes130US, Electricity. These datasets can be download from here [1]. For each dataset, we use it as in-distribution data while all the other datasets are treated as out-of-distribution data. We also generate random samples by using standard Gaussian and uniform distribution for each feature value in the tabular as out-of-distribution datasets. All the feature values of these datasets are rescaled into the range $[0, 1]$ for experiments.

**Image data**   We perform evaluation on two different types of images: gray-scale images and colored images. For the former, we use MNIST [2] and FashionMNIST (Xiao et al., 2017) as in-distribution data, use MedMNIST (Yang et al., 2023) (which is a collection of MNIST-like medical image datasets), KMNIST (Clanuwat et al., 2018), QMNIST (Yadav & Bottou, 2019), Omniglot (Lake et al., 2015), NotMNIST [3], CIFAR-10bw (Krizhevsky, 2009) as out-of-distribution data. For the latter, we use CIFAR-10 (Krizhevsky, 2009) and CIFAR-100 (Krizhevsky, 2009) as in-distribution data, SVHN (Netzer et al., 2011), Texture (Cimpoi et al., 2014), Places365 (Zhou

---

[1] https://huggingface.co/datasets/inria-soda/tabular-benchmark
[2] https://yann.lecun.com/exdb/mnist/
[3] https://www.kaggle.com/datasets/lubaroli/notmnist

et al., 2018), iSUN (Xu et al., 2015), LSUN-Crop (Yu et al., 2016), LSUN-Resize (Yu et al., 2016), STL-10 (Coates et al., 2011) as out-of-distribution data. In both cases, we also generate random images using standard Gaussian and uniform distribution for each pixel value as out-of-distribution datasets. The pixel values generated from Gaussian distribution are clipped into the range $[0, 1]$.

**Text data**   We perform evaluation on text data such as IMDB, AGNEWS, Amazon, YahooAnswers, Yelp, which can be downloaded here [4]. Each time, we use one of the text datasets as in-distribution data while all the other datasets are treated as out-of-distribution data. We also use the three categories 'Computer', 'Sports', 'Politics' from 20Newsgroups (Lang, 1995) dataset.

## B   BASELINES AND EVALUATION

We compare the model performance with the state-of-the-art OOD detection methods such as MSP (Hendrycks & Gimpel, 2017), ODIN (Liang et al., 2018), Mahalanobis (Lee et al., 2018),OE (Hendrycks et al., 2019), Energy score (Liu et al., 2020), and PnPOOD (Rawat et al., 2021).

We choose false positive rates of out-of-distribution samples when the true positive rate of in-distribution samples are at $90\%$ and $95\%$ (FPR90 and FPR95), the area under the receiver operating characteristic curve (AUROC), the area under the precision-recall curve (AUPR) as the evaluation metrics for model performance. For these metrics, $\uparrow$ indicates larger value is better, $\downarrow$ indicates smaller

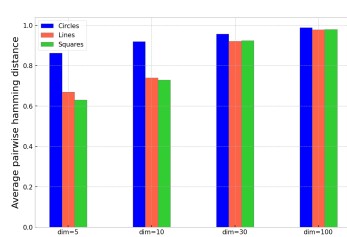

Figure 7: APHD for simulated data with different dimensions.

valuer is better. For all experiments, we use in-distribution data as positive samples and out-of-distribution data as negative samples. All experimental results shown are percentage and are averaged over 10 individual runs.

## C   FURTHER DETAILS OF EXPERIMENTAL PROCEDURE AND HYPERPARAMETERS

As summarized in Algorithm 1, we first fit a tree-based model, e.g., random forest or extremely randomized tree, by using the training part of an in-distribution dataset and then compute the APHD by feeding both the testing part of in-distribution and out-of-distribution datasets. It is worth noting that the out-of-distribution data is not used during the tree-based model fitting phase for most cases, the tree-based model is only fitted by using the training part of each individual in-distribution dataset except for the cases with CIFAR-10 and CIFAR-100.

For all experiments, we randomly choose 500 samples from in-distribution and out-of-distribution testing data each time to calculate their average pairwise hamming distance (AHPD) values, and we repeat each experiment 10 times to get 5000 APHD values. These values are used for determining the AUROC, ARPR, and FPR90 or FPR95 scores.

Our code in the supplementary material can be run by putting the code in each cell of the '.py' file sequentially into google colab. The 'TOOD' folder should be under the directory '/content/gdrive/MyDrive'. Unfortunately, we couldn't include all the datasets because of the file size limit for the submission system, but we are happy to provide all the datasets later on during the review to verify that our results are genuine.

### C.1   DATA PREPROCESSING

For tabular data, we directly use the original training samples as input for fitting the tree-based ensemble model. For image data, we apply an autoencoder with certain structure to extract latent features in the training images and use the latent features as input for tree-based model learning. For

---

[4] https://pytorch.org/text/stable/datasets.html

text data, we apply GloVe (Pennington et al., 2014) with 6B tokens and 300d. We take average over all the words embedding in each sentence and use the average embedding as the input for tree-based ensemble model.

For all experiments involved in MNIST and FashionMNIST, we use a convolutional autoencoder whose first layer of its encoder contains 1 in-channel and 16 out-channels, the kernel and padding sizes are set to be 3 and 1 respectively. The second layer contains 16 in-channel and 4 out-channels, the kernel and padding size are set to be 3 and 1 respectively. Then we apply a maximum pooling with kernel and stride size both equal to 2. For the decoder part, the first layer contains 4 in-channels and 16 out-channels, the kernel and stride size are set to be 2. The second layer contains 16 in-channels and 1 out-channel, the kernel and stride size are set to be 2. The ReLU activations are added between inner layers and sigmoid activation is added after the last layer of autoencoder. All other hyperparameters are the default values in pytorch.

For all experiments involved in CIFAR-10 and CIFAR-100, we use a convolutional autoencoder whose first layer of its encoder contains 3 in-channels and 16 out-channels, the kernel, stride, padding sizes are set to be 3, 2, 1 respectively. The second layer contains 16 in-channel and 32 out-channels, the kernel, stride, padding sizes are set to be 3, 2, 1 respectively. The third layer contains 32 in-channel and 64 out-channels, the kernel, stride, and padding sizes are set to be 3, 2, 1 respectively. For the decoder part, the first layer contains 64 in-channels and 32 out-channels, the kernel, stride, padding, output-padding sizes are set to be 3, 2, 1, 1 respectively. The second layer contains 32 in-channels and 16 out-channel, the kernel, stride, padding, output-padding sizes are set to be 3, 2, 1, 1 respectively. The third layer contains 16 in-channels and 3 out-channel, the kernel, stride, padding sizes are set to be 4, 2, 1 respectively. The ReLU activation are added between inner layers and sigmoid activation is added after the last layer of autoencoder. All other hyperparameters are the default values in pytorch.

For experiments on MNIST and FashionMNIST, we train the autoencoder with 30 epochs on the training dataset. For the experiments on CIFAR-10 and CIFAR-100, and we train the autoencoder with 3 epochs on the training dataset and 30 epochs on each of the testing datasets.

## C.2 Hyperparameters

To speed up the computation, we apply the extremely randomized tree (ExtraTree) model in all experiments instead of random forest. However, one can apply random forest model as well to get similar experimental results.

For simulated data and tabular data tasks, we set the hyperparameter $min\_samples\_leaf = 1$ in the python *sklearn.ensemble.ExtraTreesClassifier* model, while in other tasks, we fix $min\_samples\_leaf = 100$. For computer vision and natural language taskes, we set $n\_estimators = 500$, while in other tasks, we fix $n\_estimators = 100$. In all experiments, we fix $max\_features =' sqrt', bootstrap = True, class\_weight =' balance'$. All other hyperparameters are the model's default values.

From our experience, the hyperparameters used in training the tree models do not have much effect on the OOD detection results, as long as the parameters not chosen too extreme.

## D Deferred proofs

Let us show the missing proofs of lemma and theorems in Section 4.

## D.1 Proof of Lemma 1

*Proof.* We would like to show that $d(T_\ell(\boldsymbol{x}_1), T_\ell(\boldsymbol{x}_2)) = 0$ if and only if $\boldsymbol{x}_1, \boldsymbol{x}_2 \in H$ for some $H \in \mathcal{H}_\ell$.

For sufficiency, suppose we have $\boldsymbol{x}_1, \boldsymbol{x}_2 \in H$ for some $H \in \mathcal{H}_\ell$. Then $\boldsymbol{x}_1$ and $\boldsymbol{x}_2$ will follow the same decision constraints path in $\ell$-th tree until reaching the leaf node. Hence $T_\ell(\boldsymbol{x}_1) = T_\ell(\boldsymbol{x}_2)$.

For necessity, let us proceed by contradiction. Suppose otherwise, then for any $\boldsymbol{x}_1, \boldsymbol{x}_2$, there is at least an $H \in \mathcal{H}_\ell$ such that $\boldsymbol{x}_1 \in H$ but $\boldsymbol{x}_2 \notin H$. Then $\boldsymbol{x}_1, \boldsymbol{x}_2$ will be separated through a decision

node corresponding to $H$ in $\mathcal{H}_\ell$, and hence appears on different leaves in $\ell$-th tree. Therefore, we have $T_\ell(\boldsymbol{x}_1) \neq T_\ell(\boldsymbol{x}_2)$ and hence $d(T_\ell(\boldsymbol{x}_1), T_\ell(\boldsymbol{x}_2)) = 1$. $\square$

### D.2 PROOF OF THEOREM 1

*Proof.* Since we have assumed that the trees are pruned to be minimal and the number of samples for each leaf node equals to 1, it is easy to see that the decision regions will be made such that both sides of each decision boundary will contain some samples in $\mathcal{D}_{train}$. Since $Conv(supp(\mathcal{D}_{test})) \cap Conv(supp(\mathcal{D}_{train})) = \emptyset$, the samples in $\mathcal{D}_{test}$ will not cross any of the decision boundaries which obtained from the training phase. In other words, for any $\boldsymbol{x}_i, \boldsymbol{x}_j \in \mathcal{D}_{test}$, there is at least an $H \in \mathcal{H}_\ell$ such that $\boldsymbol{x}_i, \boldsymbol{x}_j \in H$. By Lemma 1, we have $d(T_\ell(\boldsymbol{x}_i), T_\ell(\boldsymbol{x}_j)) = 0$. $\square$

### D.3 PROOF OF THEOREM 2

*Proof.* If $\mathcal{D}_{test}$ is the same as $\mathcal{D}_{train}$, for any pair of samples $\boldsymbol{x}_1, \boldsymbol{x}_2 \in \mathcal{D}_{test}$, they will reach at the same leaf node if they belong to a same decision region in $\mathcal{H}_\ell$, and hence have tree embedded hamming distance 0. Otherwise, they will have tree embedded hamming distance 1. Since there are $K$ regions in total, the probability for any pair of samples which happen to be in the same decision boundary equals to $K(1/K)^2 = \frac{1}{K}$. Therefore we have $\mathbb{E}[d(T_\ell(\boldsymbol{x}_1), T_\ell(\boldsymbol{x}_2))] = 1 - \frac{1}{K} = \frac{K-1}{K}$. $\square$

### D.4 PROOF OF THEOREM 3

*Proof.* Since we have assumed the decision boundaries are orthogonal to the axes, there will be no decision boundaries intersect with the region $[b_1, b_1 + a_2 - a_1]^n$. Therefore, if both $\boldsymbol{x}_i, \boldsymbol{x}_j \in [b_1, b_1 + a_2 - a_1]^n$, their hamming distance equals to $d(\boldsymbol{x}_i, \boldsymbol{x}_j) = 0$. The probability for this case to happen equals to $\left( (\frac{a_2-a_1}{b_1-a_1})^n \right)^2 = (\frac{a_2-a_1}{b_1-a_1})^{2n}$.

If both $\boldsymbol{x}_i, \boldsymbol{x}_j \in [a_2, b_1 + a_2 - a_1]^n \setminus [a_1, b_1]^n$, since $k \gg 1$ for each dimension, they will be almost surely lie on different sides of some decision boundary, and hence their hamming distance equals to $d(\boldsymbol{x}_i, \boldsymbol{x}_j) = 1$. The probability for this case to happen equals to $(1 - (\frac{a_2-a_1}{b_1-a_1})^n)^2$.

If $\boldsymbol{x}_i \in [b_1, b_1 + a_2 - a_1]^n$, $\boldsymbol{x}_j \in [a_2, b_1 + a_2 - a_1]^n \setminus [b_1, b_1 + a_2 - a_1]^n$, or vice versa, their hamming distance also equals to $d(\boldsymbol{x}_i, \boldsymbol{x}_j) = 1$. The probability for this case to happen equals to $2 \cdot (\frac{a_2-a_1}{b_1-a_1})^n \cdot (1 - (\frac{a_2-a_1}{b_1-a_1})^n)$.

Therefore, the expected pairwise hamming distance of $\boldsymbol{x}_i, \boldsymbol{x}_j$ equals to the weighted average of above cases, which is

$$\mathbb{E}[d(T_\ell(\boldsymbol{x}_i), T_\ell(\boldsymbol{x}_j))] = 0 \cdot \left( \frac{a_2 - a_1}{b_1 - a_1} \right)^{2n} + 1 \cdot \left( 1 - \left( \frac{a_2 - a_1}{b_1 - a_1} \right)^{2n} \right) = 1 - \left( \frac{a_2 - a_1}{b_1 - a_1} \right)^{2n}. \quad (8)$$

$\square$

### D.5 PROOF OF THEOREM 4

*Proof.* The first result follows directly since $\mathbb{E}[d(T_\ell(\boldsymbol{x}_i), T_\ell(\boldsymbol{x}_j))] = 1 - (\frac{a_2-a_1}{b_1-a_1})^{2n}$ for $\ell = 1, \cdots, L$. The second result follows since $d(T(\boldsymbol{x}_i), T(\boldsymbol{x}_j)) = \frac{1}{L} \sum_{\ell=1}^L d(T_\ell(\boldsymbol{x}_i), T_\ell(\boldsymbol{x}_j))$, and we can apply Hoeffding's inequality with $\mathbb{E}[d(T_\ell(\boldsymbol{x}_i), T_\ell(\boldsymbol{x}_j))] = 1 - (\frac{a_2-a_1}{b_1-a_1})^{2n}$ for $\ell = 1, \cdots, L$. $\square$

## E EXPANDED EXPERIMENTAL RESULTS

Table 7: Expanded TOOD detection results for MNIST. The standard deviation values smaller than 1% are omitted.

| $\mathcal{D}_{in}$ | $\mathcal{D}_{out}$ | AUROC ↑ | AUPR ↑ | FPR95 ↓ |
|---|---|---|---|---|
| **MNIST** | BreastMNIST | 100 | 100 | 0 |
| | ChestMNIST | 100 | 100 | 0 |
| | OctMNIST | 100 | 100 | 0 |
| | OrganaMNIST | 99.9 | 99.9 | 0.4 |
| | OrgancMNIST | 99.8 | 99.7 | 0.6 |
| | OrgansMNIST | 99.8 | 99.8 | 0.6 |
| | PneumMNIST | 100 | 100 | 0 |
| | TissueMNIST | 100 | 100 | 0 |
| | KMNIST | $95.8 \pm 2.32$ | $96.7 \pm 2.17$ | $20.2 \pm 3.65$ |
| | QMNIST | $48.7 \pm 9.8$ | $49.6 \pm 8.6$ | $96.4 \pm 2.98$ |
| | Omniglot | 100 | 100 | 0 |
| | CIFAR-10bw | 100 | 100 | 0 |
| | NotMNIST | 100 | 100 | 0 |
| | Gaussian | 100 | 100 | 0 |
| | Uniform | 100 | 100 | 0 |

Table 8: Expanded TOOD detection results for FashionMNIST. The standard deviation values smaller than 1% are omitted.

| $\mathcal{D}_{in}$ | $\mathcal{D}_{out}$ | AUROC ↑ | AUPR ↑ | FPR95 ↓ |
|---|---|---|---|---|
| **FashionMNIST** | BreastMNIST | 100 | 100 | 0 |
| | ChestMNIST | 99.9 | 99.8 | 0.1 |
| | OctMNIST | 100 | 100 | 0 |
| | OrganaMNIST | 99.9 | 99.9 | 0.4 |
| | OrgancMNIST | 99.6 | 99.2 | 1.1 |
| | OrgansMNIST | 99.5 | 98.5 | 1.0 |
| | PneumMNIST | 100 | 100 | 0 |
| | TissueMNIST | 100 | 100 | 0 |
| | KMNIST | 99.2 | 99.1 | $4.2 \pm 1.12$ |
| | QMNIST | 100 | 100 | 0 |
| | Omniglot | 100 | 100 | 0 |
| | CIFAR-10bw | 99.6 | $98.0 \pm 1.36$ | 0.5 |
| | NotMNIST | 100 | 100 | 0 |
| | Gaussian | 100 | 100 | 0 |
| | Uniform | 100 | 100 | 0 |

Table 9: Expanded TOOD detection results for MNIST after random shuffling of labels. The standard deviation values smaller than $1\%$ are omitted.

| $\mathcal{D}_{in}$ | $\mathcal{D}_{out}$ | AUROC ↑ | AUPR ↑ | FPR95 ↓ |
|---|---|---|---|---|
| | BreastMNIST | 100 | 100 | 0 |
| | ChestMNIST | 100 | 100 | 0 |
| | OctMNIST | 100 | 100 | 0 |
| | OrganaMNIST | 100 | 100 | 0.1 |
| | OrgancMNIST | 99.8 | 99.7 | 1.2 |
| | OrgansMNIST | 99.9 | 99.9 | 0.2 |
| | PneumMNIST | 100 | 100 | 0 |
| MNIST | TissueMNIST | 100 | 100 | 0 |
| | KMNIST | $93.2 \pm 3.32$ | $95.0 \pm 2.68$ | $52.0 \pm 10.73$ |
| | QMNIST | $48.7 \pm 9.6$ | $49.6 \pm 8.5$ | $97.0 \pm 1.46$ |
| | Omniglot | 100 | 100 | 0 |
| | CIFAR-10bw | 100 | 100 | 0 |
| | NotMNIST | 100 | 100 | 0 |
| | Gaussian | 100 | 100 | 0 |
| | Uniform | 100 | 100 | 0 |

Table 10: Expanded TOOD detection results for FashionMNIST after random shuffling of labels. The standard deviation values smaller than $1\%$ are omitted.

| $\mathcal{D}_{in}$ | $\mathcal{D}_{out}$ | AUROC ↑ | AUPR ↑ | FPR95 ↓ |
|---|---|---|---|---|
| | BreastMNIST | 100 | 100 | 0 |
| | ChestMNIST | 99.8 | 99.0 | 0.3 |
| | OctMNIST | 100 | 100 | 0 |
| | OrganaMNIST | 100 | 100 | 0.1 |
| | OrgancMNIST | 99.8 | 99.5 | 1.1 |
| | OrgansMNIST | 99.9 | 99.9 | 0.4 |
| | PneumMNIST | 100 | 100 | 0 |
| FashionMNIST | TissueMNIST | 100 | 100 | 0 |
| | KMNIST | $97.8 \pm 1.87$ | $98.0 \pm 1.23$ | $13.2 \pm 4.38$ |
| | QMNIST | 100 | 100 | 0 |
| | Omniglot | 100 | 100 | 0 |
| | CIFAR-10bw | 99.9 | 99.9 | 0.5 |
| | NotMNIST | 100 | 100 | 0 |
| | Gaussian | 100 | 100 | 0 |
| | Uniform | 100 | 100 | 0 |

Table 11: Expanded TOOD detection results on text data. The standard deviation values smaller than 1% are omitted.

| $\mathcal{D}_{in}$ | $\mathcal{D}_{out}$ | AUROC ↑ | AUPR ↑ | FPR95 ↓ |
|---|---|---|---|---|
| IMDB | AGNEWs | 100 | 100 | 0 |
| | Amazon | $96.2 \pm 1.56$ | $93.6 \pm 2.14$ | $8.61 \pm 3.21$ |
| | YahooAnswers | 99.4 | 98.9 | 2.02 |
| | Yelp | 99.8 | 99.5 | 0.61 |
| AGNEWS | IMDB | 100 | 100 | 0 |
| | Amazon | 99.7 | 98.7 | 0.62 |
| | YahooAnswers | 98.8 | $96.7 \pm 1.82$ | $3.98 \pm 1.93$ |
| | Yelp | 100 | 100 | 0 |
| Amazon | IMDB | 98.3 | $94.3 \pm 2.31$ | $2.62 \pm 1.42$ |
| | AGNEWS | 99.5 | 98.3 | 0.62 |
| | YahooAnswers | $97.4 \pm 1.54$ | $93.6 \pm 2.76$ | $7.03 \pm 3.42$ |
| | Yelp | 98.6 | $95.8 \pm 1.65$ | $2.41 \pm 1.12$ |
| YahooAnswers | IMDB | 99.6 | 98.9 | 1.04 |
| | AGNEWS | 98.9 | 97.6 | 1.78 |
| | Amazon | $93.7 \pm 2.54$ | $90.2 \pm 3.62$ | $19.4 \pm 5.87$ |
| | Yelp | 98.0 | $95.4 \pm 1.42$ | $4.43 \pm 1.92$ |
| Yelp | IMDB | 99.8 | 99.7 | 0.61 |
| | AGNEWS | 100 | 100 | 0 |
| | Amazon | 98.6 | $97.7 \pm 1.29$ | $4.20 \pm 1.35$ |
| | YahooAnswers | 99.5 | 99.3 | 1.81 |

Table 12: Expanded TOOD detection results for CIFAR-10. The standard deviation values smaller than 1% are omitted.

| $\mathcal{D}_{in}$ | $\mathcal{D}_{out}$ | AUROC ↑ | AUPR ↑ | FPR95 ↓ |
|---|---|---|---|---|
| CIFAR-10 | SVHN | 99.4 | 98.9 | 2.32 |
| | Texture | 99.5 | 98.8 | 1.74 |
| | Places365 | $97.6 \pm 1.65$ | $97.2 \pm 1.34$ | $10.9 \pm 3.76$ |
| | iSUN | 98.6 | 98.2 | $5.85 \pm 2.15$ |
| | LSUN-Crop | 99.9 | 99.8 | 0.74 |
| | LSUN-Resize | 99.6 | 99.6 | 1.94 |
| | STL-10 | $83.2 \pm 5.23$ | $78.6 \pm 7.82$ | $46.2 \pm 12.31$ |

Table 13: Expanded TOOD detection results for CIFAR-100. The standard deviation values smaller than 1% are omitted.

| $\mathcal{D}_{in}$ | $\mathcal{D}_{out}$ | AUROC ↑ | AUPR ↑ | FPR95 ↓ |
|---|---|---|---|---|
| CIFAR-100 | SVHN | $96.4 \pm 1.91$ | $95.7 \pm 1.97$ | $19.5 \pm 5.27$ |
| | Texture | 99.1 | 98.7 | $4.30 \pm 1.62$ |
| | Places365 | $87.9 \pm 3.48$ | $90.1 \pm 2.73$ | $76.6 \pm 7.93$ |
| | iSUN | $88.8 \pm 4.23$ | $88.8 \pm 4.19$ | $54.8 \pm 8.92$ |
| | LSUN-Crop | 98.8 | 98.6 | $8.02 \pm 2.52$ |
| | LSUN-Resize | $93.9 \pm 2.32$ | $94.2 \pm 2.19$ | $37.3 \pm 5.91$ |
| | STL-10 | $75.0 \pm 6.37$ | $67.9 \pm 8.83$ | $62.4 \pm 9.31$ |

Table 14: Expanded TOOD detection results for CIFAR-10 after random shuffling of labels. The standard deviation values smaller than 1% are omitted.

| $\mathcal{D}_{in}$ | $\mathcal{D}_{out}$ | AUROC ↑ | AUPR ↑ | FPR95 ↓ |
|---|---|---|---|---|
| **CIFAR-10** | SVHN | 99.5 | 99.1 | 2.15 |
| | Texture | 99.4 | 99.3 | 2.62 |
| | Places365 | $97.1 \pm 1.34$ | $97.2 \pm 1.18$ | $16.0 \pm 5.61$ |
| | iSUN | $97.2 \pm 1.72$ | $95.9 \pm 2.18$ | $12.9 \pm 3.71$ |
| | LSUN-Crop | 99.9 | 99.9 | 0.13 |
| | LSUN-Resize | 99.2 | 98.9 | $3.53 \pm 1.75$ |
| | STL-10 | $53.9 \pm 8.45$ | $49.2 \pm 11.63$ | $76.7 \pm 8.68$ |

Table 15: Expanded TOOD detection results for CIFAR-100 after random shuffling of labels. The standard deviation values smaller than 1% are omitted.

| $\mathcal{D}_{in}$ | $\mathcal{D}_{out}$ | AUROC ↑ | AUPR ↑ | FPR95 ↓ |
|---|---|---|---|---|
| **CIFAR-100** | SVHN | $94.3 \pm 2.51$ | $94.5 \pm 2.32$ | $28.6 \pm 7.62$ |
| | Texture | 99.8 | 99.8 | 1.05 |
| | Places365 | $84.2 \pm 4.72$ | $87.3 \pm 4.23$ | $71.7 \pm 6.71$ |
| | iSUN | $88.3 \pm 3.81$ | $89.3 \pm 3.62$ | $41.8 \pm 6.86$ |
| | LSUN-Crop | 99.1 | 99.1 | $5.00 \pm 2.43$ |
| | LSUN-Resize | $90.5 \pm 2.86$ | $91.1 \pm 2.59$ | $37.8 \pm 4.37$ |
| | STL-10 | $68.6 \pm 6.82$ | $64.5 \pm 7.47$ | $63.3 \pm 5.49$ |

Table 16: Expanded TOOD detection results with FGSM attack for CIFAR-10. The standard deviation values smaller than 1% are omitted.

| $\mathcal{D}_{in}$ | $\mathcal{D}_{out}$ | AUROC ↑ | AUPR ↑ | FPR95 ↓ |
|---|---|---|---|---|
| **CIFAR-10** | SVHN | $96.6 \pm 1.73$ | $95.7 \pm 1.98$ | $15.2 \pm 3.52$ |
| | Texture | 99.5 | 99.4 | $2.95 \pm 1.32$ |
| | Places365 | $91.1 \pm 3.81$ | $91.9 \pm 3.65$ | $44.4 \pm 6.42$ |
| | iSUN | $89.9 \pm 2.71$ | $87.9 \pm 3.21$ | $36.1 \pm 5.62$ |
| | LSUN-Crop | 99.7 | 99.6 | 1.60 |
| | LSUN-Resize | 98.4 | 98.4 | $9.72 \pm 2.31$ |
| | STL-10 | $69.0 \pm 7.89$ | $66.2 \pm 8.91$ | $73.2 \pm 6.36$ |

Table 17: Expanded TOOD detection results with FGSM attack for CIFAR-100. The standard deviation values smaller than 1% are omitted.

| $\mathcal{D}_{in}$ | $\mathcal{D}_{out}$ | AUROC ↑ | AUPR ↑ | FPR95 ↓ |
|---|---|---|---|---|
| **CIFAR-100** | SVHN | $96.5 \pm 1.93$ | $95.5 \pm 2.51$ | $12.8 \pm 2.71$ |
| | Texture | $98.5 \pm 1.07$ | $97.9 \pm 1.02$ | $5.55 \pm 2.63$ |
| | Places365 | $83.9 \pm 4.82$ | $84.0 \pm 3.92$ | $56.2 \pm 7.92$ |
| | iSUN | $82.9 \pm 3.71$ | $77.9 \pm 4.31$ | $47.2 \pm 6.82$ |
| | LSUN-Crop | $97.8 \pm 1.52$ | $97.4 \pm 1.53$ | $10.0 \pm 3.65$ |
| | LSUN-Resize | $86.2 \pm 3.81$ | $83.8 \pm 3.91$ | $45.1 \pm 6.82$ |
| | STL-10 | $68.9 \pm 6.87$ | $62.7 \pm 7.12$ | $58.2 \pm 8.52$ |

