# OpenReview forum: "Tree-based Ensemble Learning for Out-of-distribution Detection"
_ICLR.cc/2024/Conference — Submitted to ICLR 2024_

### Official Review · Reviewer_A8Ja · 2023-10-28

**Soundness:** 2 fair
**Presentation:** 2 fair
**Contribution:** 3 good
**Rating:** 5
**Confidence:** 2

**Summary:**

The paper focuses on a method called TOOD detection, which aims to improve out-of-distribution (OOD) detection in tree-based machine learning models. The paper evaluates this method on various types of data, including tabular, image, and text data. It also compares TOOD detection with existing state-of-the-art OOD detection methods and claims to show favorable or comparable performance. The paper includes mathematical validation to support its methodology and presents preliminary results that indicate the effectiveness of TOOD detection.

**Strengths:**

The paper is comprehensive and covers several types of data, making it widely applicable.
It provides rigorous theoretical results to mathematically validate its model.
The paper compares its method to existing diverse techniques, providing a benchmark for its effectiveness.
Preliminary results are promising, indicating the potential impact of the research.
The paper addresses the issue of OOD detection based on the new approach, tree-based algorithms, which is a significant problem in machine learning.

**Weaknesses:**

It only compares their models with kernel-based baselines for efficiency analysis. More comprehensive comparisons are necessary.

The "Comparison with State-of-the-Arts" section does not provide the comparison between TOOD and the recent models.

As shown in Figure 7, the proposed model may not be effective in high-dimensional cases.

**Questions:**

Please see the Weakness and Strengths sections.

---

> ### Author Response · Authors · 2023-11-17
> **Rebuttal by Authors**
>
> We want to thank the reviewer for pointing out the strength and weakness of our paper ! Here are a few responses from us.
>
> The reviewer mentioned the high dimensionality issue, and we absolutely agree. In principle, because of the curse of dimensionality, we believe so far there is no universal way to solve it.  However, one way to alleviate it is to use autoencoder to extract latent features and use latent features as input for our method, and we have applied the autoencoder in our implementation for image and text data tasks. This was commented in Remark 2 in the paper.
>
> For the literature, we thought we have compared with most of other recent OOD detection methods to the best our knowledge, would the reviewer mind mentioning specifically about which model(s) we are missing ?  Thank you for the time again !

---

### Official Review · Reviewer_jBcD · 2023-10-29

**Soundness:** 3 good
**Presentation:** 3 good
**Contribution:** 1 poor
**Rating:** 3
**Confidence:** 3

**Summary:**

This paper presents a tree-based ensemble learning approach for out-of-distribution (OOD) detection. The method involves calculating the Hamming distance for the tree embeddings obtained from a random forest trained on in-distribution data. The authors offer a comprehensive theoretical analysis of this method to substantiate their proposal. Additionally, empirical experiments are conducted on synthetic datasets and benchmark datasets to validate its effectiveness.

**Strengths:**

The strengths of this paper can be summarized as follows:

1. **Experimental Results:** The experiments demonstrated the proposal's strong performance across various synthetic and benchmark datasets.

2. **Clarity and Presentation:** The paper is meticulously structured, and the ideas presented are easily comprehensible, ensuring accessibility for readers.

**Weaknesses:**

While this paper exhibits several strengths, it also presents several weaknesses, which are outlined as follows:

1. **Limitation 1**: The idea that "out-of-distribution data may exhibit smaller Hamming distances among themselves" hinges on the assumption that the support of training and testing distributions does not overlap in each dimension. However, this assumption raises doubts as it prohibits anomalies from occurring in only one dimension.

2. **Limitation 2**: The central idea appears to implicitly assume that labels are distributed uniformly across different classes. Consider a binary classification scenario where the major class has a significantly higher probability than the minor class, and the labels are determined by whether $x_{i} < s_{i}$. In such cases, the Hamming distance of the embedding of in-distribution test data may be small among them, primarily because most of the samples reside in the same leaf as the major class.

3. **Regarding Experiments**: It is advisable to present the experimental results in the format "mean ± std" due to the inherent randomness of random forests.

4. **Regarding Related Work**: Decision tree learning and random forests should be traced back to the works of Quinlan (1979), Breiman et al. (1984), and Breiman (2001).

References:
- Quinlan JR (1979) "Discovering Rules by Induction from Large Collections of Examples." Expert Systems in the Micro Electronics Age.
- Breiman L, Friedman JH, Olshen RA, et al (1984) "Classification and Regression Trees." Chapman & Hall/CRC.
- Breiman L (2001) "Random Forests." Machine Learning, 45(1), 5–32.

**Questions:**

See Weaknesses #1 and #2.

---

> ### Author Response · Authors · 2023-11-17
> **Rebuttal by Authors**
>
> We would like to thank the reviewer for the suggestions to improve our paper ! Here are a few responses from us.
>
> The reviewer mentioned Limitation 1,  but I’m not sure if I understand the issue. We assumed in Theorem 1 that the support of training and testing distributions does not overlap in each dimension, which means along all the dimensions, the two distributions do not overlap. We have rewritten the statement in Theorem 1 and hopefully it can address the confusion.
>
> The reviewer also mentioned that the label distribution could be an issue for the tree-based. This is absolutely a great point. From our point of view, this issue can be potentially solved by manually redistributed the labels. For example, if the major class has 100 samples and the minor class has 10 samples, then we can randomly relabel such that there are 55 samples for each class.  By doing this, we will no longer have the issue that the hamming distance will be small. This idea is similar as label shuffling which we have discussed in the experiments section.
>
> We highly appreciate the reviewer for pointing out the misuse of the reference for decision tree and random forest, it is our carelessness for not making it correct in the first place, we have adapted these changes in the revised version of our paper. We also added in the standard deviation for the experimental results in the appendix.

---

> > ### Comment · Reviewer_jBcD · 2023-11-18
> > **Response to Authors' Rebuttal and Further Clarification**
> >
> > I appreciate your detailed response to my comments on your paper. I would like to further elaborate on my concern regarding Limitation 1 to ensure clarity.
> >
> > In Theorem 1, you assume that the support of training and testing distributions does not overlap in each dimension. To illustrate the concern, let's consider a scenario where the support of in-distribution data is defined as $[0, 1]^2$ with negative samples distributed in $[0, 0.5) \times [0, 0.5) \cup [0.5, 1] \times [0.5, 1]$ and positive samples distributed in $[0, 0.5) \times [0.5, 1] \cup [0.5, 1] \times [0, 0.5)$. The out-distribution data is distributed across four regions: $[-\infty, -100] \times [0, 1]$, $[0, 1] \times [100, \infty]$, $[0, 1] \times [-\infty, -100]$, and $[100, \infty] \times [0, 1]$ (It is reasonable to classify these four regions as out-distribution data).
> >
> > In this case, the proposed TOOD can fail as out-distribution data may exhibit larger Hamming distances among themselves. The constraint imposed by Theorem 1 prohibits the occurrence of anomalies in only one dimension, which may not fully align with real-world scenarios where anomalies could manifest in specific dimensions.
> >
> > I believe it is reasonable to consider anomalies in only some dimensions, and the current formulation of Theorem 1 may be overly restrictive in this regard. Consequently, I maintain my assessment of Limitation 1.

---

> > > ### Author Response · Authors · 2023-11-18
> > > **Response to Reviewer's Comment**
> > >
> > > We appreciate the reviewer's further elaboration on Limitation 1. We absolutely agree such scenario can happen in practice, and we didn't think carefully before about the cases when there are disconnection within the data distribution.
> > >
> > > In order to get the desired result in Theorem 1, the assumption that there is no overlap between the support of training and testing distribution is not enough, we need stronger assumption such as the convex hull of the support of the training and testing distribution don't intersect. It is certainly a very strong assumption, but it can be a good result to show the ideal case when training and testing distribution can be easily separated. For the general case (when there is overlap between training and testing distribution), we have addressed it as Theorem 3 in the paper.
> > >
> > > We have updated this part in the paper and I hope it can address the concern here.  Thank you again for the comments !

---

### Official Review · Reviewer_5AJZ · 2023-10-31

**Soundness:** 4 excellent
**Presentation:** 3 good
**Contribution:** 3 good
**Rating:** 8
**Confidence:** 3

**Summary:**

This study presents a novel scoring function for mismatch detection. For each test input, the position of its terminal node in a set of classification trees is recorded. Then the authors use the hamming distance between two location vectors to quantify the similarity of two sample points.

**Strengths:**

- The proposed detection method is novel and interesting.
- There is diversity in the experimental setup, considering OOD detection on multiple data types.
- Theoretical analysis is provided.

**Weaknesses:**

- This method is not valid for high-dimensional inputs.
- There are no experiments on ImageNet benchmark.
- The results of the theoretical analysis are for a single classification tree model, not for a random forest.

**Questions:**

1. For image classification, does the input $x$ refer to an image or a feature vector obtained from a pre-trained feature extractor?
2. Do the hyper-parameters used in training the tree models (such as tree depth, the number of terminal nodes, and the minimal size of terminal nodes) have any effect on OOD Detection results?

---

> ### Author Response · Authors · 2023-11-17
> **Rebuttal by Authors**
>
> We want to thank reviewer for the positive comments ! Here are a few responses from us.
>
> We certainly agree that the high-dimensional input issue is a big challenge for our method, as well as for other learning methods. In principle, because of the curse of dimensionality, we believe so far there is no universal way to solve it.  However, one way to alleviate it is to use autoencoder to extract latent features and use latent features as input for our method, and we have applied the autoencoder in our implementation for image and text data tasks. This was commented in Remark 2 in the paper.
>
> The reviewer also mentioned that the theoretical analysis are for a single classification tree model, not for a random forest. This is another great point. We have added a theorem (Theorem 4 in the updated version) to address this. So far we haven’t perform the experiments on ImageNet benchmark, and we would like to add in more experiments in the final version of our paper.
>
> For input x in image classification, it refers to the latent feature vector obtained from an autoencoder.  From our experience when conducting the experiments, the hyperparameters used in training the tree models do not have much effect on the OOD detection results (as long as the parameters not too extreme). We have added these comments in the appendix of the updated version of our paper.

---

### Official Review · Reviewer_FZKS · 2023-11-01

**Soundness:** 2 fair
**Presentation:** 2 fair
**Contribution:** 2 fair
**Rating:** 3
**Confidence:** 3

**Summary:**

The paper addresses the fundamental question of determining whether testing samples have a similar distribution to training samples, which is crucial for the safe deployment of machine learning models. The authors propose a mechanism called TOOD detection, which is a simple and effective tree-based method for detecting out-of-distribution (TOOD) samples. The TOOD detection mechanism works by computing the pairwise hamming distance of tree embeddings of the testing samples. These embeddings are obtained by fitting a tree-based ensemble model using in-distribution training samples. The authors highlight that their approach is interpretable and robust due to its tree-based nature. Additionally, the method is efficient, flexible across various machine learning tasks, and can be applied to unsupervised settings. The paper presents extensive experiments to demonstrate the superiority of the proposed method compared to other state-of-the-art out-of-distribution detection methods. The experiments cover tabular, image, and text data, showing the effectiveness of the approach in distinguishing between in-distribution and out-of-distribution samples.

**Strengths:**

Novel approach: The paper introduces a new mechanism, TOOD detection, which offers a novel perspective on addressing the problem of determining whether testing samples have a similar distribution to training samples by a tree based ensemble method. From my personal knowledge, tree structures and ensemble methods are seldomly studied in OOD detection, making the considered direction an interesting line of works.


Effective methodology: The proposed TOOD detection mechanism based on computing pairwise hamming distances of tree embeddings proves to be simple yet effective in distinguishing in-distribution from out-of-distribution samples. The approach demonstrates superior performance compared to other state-of-the-art methods in extensive experiments across various data types.

Interpretable and robust: The authors highlight the interpretability and robustness of their approach, attributed to its tree-based nature. This characteristic allows for better understanding and trust in the detection process, making it easier to analyze and interpret the results.

Flexibility across machine learning tasks: The paper emphasizes the flexibility of the proposed approach, indicating that it can be applied to various machine learning tasks. This versatility makes it applicable to a wide range of scenarios, adding practical value to the research.

Generalizability to unsupervised setting: The authors state that their method can be easily generalized to unsupervised settings, which is beneficial in scenarios where labeled data is scarce or unavailable. This adaptability enhances the applicability of the proposed approach.

**Weaknesses:**

The authors define OOD in the abstract, but such a definition may violate the main stream of the community. In my view, telling the difference between two distributions is more related to two sample test. While in OOD Detection,  we typically assume the ID and OOD distribution has been mixed, thus we need to tell data as ID and OOD cases instance/point wise. I think such a setting is more difficult than two sample test, making OOD detection remain a challenging task in the literature. It will be great if the authors can discuss about it.

The paper is not clearly written. I am not sure if the proposed tree based method uses original features in the input space or embedding feature given by the pretrained classifier. If the former is true, I am not sure if the tree based methods have enough capability to fit complex classification tasks such as CIFAR classification. Also, the computational complexity will be high (even built upon the high dimensional embedding features). If the latter is true, I am not sure if the learned embedding features are good enough in OOD detection, especially considering the reliance of strong assumptions in their theoretical analysis (see also in the below questions).

A related question is about the strong assumption in Theorem 1. In the input space, especially for the complex image classification task, it is obviously not true. In the embedding space, since model cannot perfectly separate ID and OOD cases, it is still a strong assumption. Therefore, I cannot fully understand why the tree based method is superior over previous works such as distance based methods (KNN), MSP, Energy, among many others.

Why ensemble method can facilitate OOD detection, the theoretical analysis does not cover such an issue, meanwhile heuristic explanation and empirical evaluation  are not sufficient. Therefore, I think the authors should discuss more about why ensembling is critical for the suggested tree based methods.

The authors make another strong assumption that the calibration failures, which is the main cause of why DNNs fail in OOD detection, will not occur for tree based methods. I am not sure if it is true in the real world, and more evaluation and ablation should be provided.

More discussion about the hyper parameter setting and the choice of evaluation datasets should be discussed here. More experiments about hard OOD detection and wild OOD detection are also of the interest in the literature.

**Questions:**

Please see the Weaknesses above.

---

> ### Author Response · Authors · 2023-11-17
> **Rebuttal by Authors**
>
> We would like to thank the reviewer for very detailed comments ! Here are a few responses from us.
>
> The reviewer mentioned the instance/point-wise test. This is a great point and we actually did such test as well when we were writing the paper. One way to perform instance-based test by using tree-based method is that we can use each instance as the center of some Gaussian distribution, with the variance similar to the original dataset, to generate a set of fake samples. In this way, we will have many samples, then we can apply the same method as described in this paper. We choose to exclude the instance based test mainly because that the other algorithms we compare with do not use instance-based test. We have added another result (Theorem 4 in the updated version) to address that the instance-based test can be done by using our method.
>
> For the issue of whether the tree-based method uses original features or embedded features as input, we mentioned at the beginning of Section 5 in the paper: “For simulated and tabular data, we directly use the original training samples as input for the tree-based model. For image or text data, we impose an autoencoder or word embedding to extract latent features and use latent features as input for our tree-based model.” Intuitively, we should directly use the original features if the dataset is simple, and use embedded features if the dataset is complicated. In terms of the computational cost, if we use algorithm such as random forest, then the cost will be high as each node splitting has to be optimized. However, in our implementation, we use the Extremely Randomized Tree (ExtraTree), whose node splitting is completely random, and it can also achieve the same performance as random forest.
>
> For Theorem 1, we agree it has a very strong assumption. However, this is just the our first theorem to address the ideal case. We addressed the general case (when there are overlaps between training and testing dataset) as Theorem 3 in the paper.
>
> The reviewer mentioned about why ensemble method can facilitate OOD detection. This is another great point. We have added a theorem (Theorem 4 in the updated version) to address this.
>
> For the calibration failures, I’m not sure what is it referring to exactly, would you mind giving some more explanations here ?  For the hyperparameters setting and evaluation datasets, they are discussed in the appendix.

---

### Author Response · Authors · 2023-11-17
**Author Response**

Dear reviewers,

We are grateful for all the suggestions made from you, and we believe most of the concerns have been addressed in our rebuttal comments below. The main changes we have made in our updated version of the paper is we added in the consistency result (as Theorem 4) of pairwise hamming distance for ensemble of multiple trees.

We also addressed the high dimensionality issue by using feature extractor such as autoencoder to reduce the dimensionality of input. Although for complicated high dimensional dataset, the features being extracted in this way may not be very good for classification tasks if using tree-based model, they are actually quite good for OOD detection tasks with tree-based model.  Some other minor changes are also being made through the paper.

We would like to ask the reviewers to take another look of our updated version and we will appreciate your time for making any future comments. Thank you again for all your valuable suggestions !


Best,

Authors

---

### Meta-Review · Area_Chair_5TuE · 2023-12-05

**Metareview:**

The paper addresses the fundamental question of determining whether testing samples have a similar distribution to training samples, which is crucial for the safe deployment of machine learning models. The authors propose a mechanism called TOOD detection, which is a simple and effective tree-based method for detecting out-of-distribution (TOOD) samples. The TOOD detection mechanism works by computing the pairwise hamming distance of tree embeddings of the testing samples. These embeddings are obtained by fitting a tree-based ensemble model using in-distribution training samples. The authors highlight that their approach is interpretable and robust due to its tree-based nature. Additionally, the method is efficient, flexible across various machine learning tasks, and can be applied to unsupervised settings. The paper presents extensive experiments to demonstrate the superiority of the proposed method compared to other state-of-the-art out-of-distribution detection methods. The experiments cover tabular, image, and text data, showing the effectiveness of the approach in distinguishing between in-distribution and out-of-distribution samples. Specifically, the strength of this paper includes several aspects. 1) Experimental Results: The experiments demonstrated the proposal's strong performance across various synthetic and benchmark datasets. 2) Clarity and Presentation: The paper is meticulously structured, and the ideas presented are easily comprehensible, ensuring accessibility for readers.

However, there are several points to be further improved. For example, The authors define OOD in the abstract, but such a definition may violate the main stream of the community. A related question is about the strong assumption in Theorem 1. In the input space, especially for the complex image classification task, it is obviously not true. The idea that "out-of-distribution data may exhibit smaller Hamming distances among themselves" hinges on the assumption that the support of training and testing distributions does not overlap in each dimension. However, this assumption raises doubts as it prohibits anomalies from occurring in only one dimension. The central idea appears to implicitly assume that labels are distributed uniformly across different classes. Consider a binary classification scenario where the major class has a significantly higher probability than the minor class, and the labels are determined by whether. In such cases, the Hamming distance of the embedding of in-distribution test data may be small among them, primarily because most of the samples reside in the same leaf as the major class. It is advisable to present the experimental results in the format "mean ± std" due to the inherent randomness of random forests. More discussion about the hyper parameter setting and the choice of evaluation datasets should be discussed here. Therefore, this paper cannot be accepted at ICLR this time, but the enhanced version is highly encouraged to submit other top-tier venues.

**Justification For Why Not Higher Score:**

However, there are several points to be further improved. For example, The authors define OOD in the abstract, but such a definition may violate the main stream of the community. A related question is about the strong assumption in Theorem 1. In the input space, especially for the complex image classification task, it is obviously not true. The idea that "out-of-distribution data may exhibit smaller Hamming distances among themselves" hinges on the assumption that the support of training and testing distributions does not overlap in each dimension. However, this assumption raises doubts as it prohibits anomalies from occurring in only one dimension. The central idea appears to implicitly assume that labels are distributed uniformly across different classes. Consider a binary classification scenario where the major class has a significantly higher probability than the minor class, and the labels are determined by whether. In such cases, the Hamming distance of the embedding of in-distribution test data may be small among them, primarily because most of the samples reside in the same leaf as the major class. It is advisable to present the experimental results in the format "mean ± std" due to the inherent randomness of random forests. More discussion about the hyper parameter setting and the choice of evaluation datasets should be discussed here. Therefore, this paper cannot be accepted at ICLR this time, but the enhanced version is highly encouraged to submit other top-tier venues.

**Justification For Why Not Lower Score:**

However, there are several points to be further improved. For example, The authors define OOD in the abstract, but such a definition may violate the main stream of the community. A related question is about the strong assumption in Theorem 1. In the input space, especially for the complex image classification task, it is obviously not true. The idea that "out-of-distribution data may exhibit smaller Hamming distances among themselves" hinges on the assumption that the support of training and testing distributions does not overlap in each dimension. However, this assumption raises doubts as it prohibits anomalies from occurring in only one dimension. The central idea appears to implicitly assume that labels are distributed uniformly across different classes. Consider a binary classification scenario where the major class has a significantly higher probability than the minor class, and the labels are determined by whether. In such cases, the Hamming distance of the embedding of in-distribution test data may be small among them, primarily because most of the samples reside in the same leaf as the major class. It is advisable to present the experimental results in the format "mean ± std" due to the inherent randomness of random forests. More discussion about the hyper parameter setting and the choice of evaluation datasets should be discussed here. Therefore, this paper cannot be accepted at ICLR this time, but the enhanced version is highly encouraged to submit other top-tier venues.

---

### Decision · Program_Chairs · 2024-01-16

Reject